# AV-Odyssey Bench: From Fundamental Audio Perception to Audio-Visual Understanding

## Abstract

Recent multimodal large language models (MLLMs), such as GPT-4o, Gemini 1.5/2.5 Pro, and Reka Core, have advanced audio-visual reasoning capabilities, achieving strong performance in tasks like cross-modal understanding and generation. However, our **DeafTest** uncovers unanticipated failures: most of the state-of-the-art MLLMs struggle with very simple audio tasks, such as *distinguishing louder sounds* or *sound counting*. This raises a fundamental question—does a deficiency in low-level audio perception constrain higher-level audio-visual reasoning? To address this, we introduce **AV-Odyssey Bench**—a comprehensive benchmark of 4,555 meticulously designed problems that integrate text, audio, and visual modalities. Each task requires models to unify cross-modal reasoning, leveraging synchronized audio-visual cues to infer solutions. By structuring questions as multiple-choice, we ensure objective, reproducible evaluations without reliance on subjective human or LLM-based judgments. Through comprehensive benchmarking of closed-source and open-source models, we showcase: (i) current MLLMs lack robust audio-visual integration ability and (ii) performance on Deaf-Test (Pearson's $r = 0.945$) strongly correlates with AV-Odyssey accuracy. These findings not only challenge prevailing assumptions about the "multimodal proficiency" of leading models, but also highlight the importance of fundamental audio perception as a bottleneck for audio-visual reasoning. We believe that our results provide concrete guidance for future research in future dataset design, alignment strategies, and architectures toward truly integrated audio-visual understanding.

## 1 Introduction

Multimodal reasoning has advanced significantly through two key stages: vision-language models (VLMs) and their evolution into audio-visual extensions. Early VLMs, such as GPT-4V(ision) (OpenAI, 2023), pioneered visual perception capabilities, enabling tasks like object counting (Xu et al., 2023), numerical reasoning on tabular data (Yang et al., 2023), and geometric problem-solving (Zhang et al., 2025). Building on this foundation, modern Multimodal Large Language Models (MLLMs)[1] integrate *audio* modalities, exemplified by GPT-4o (Hurst et al., 2024), Gemini 1.5 (Team et al., 2024a), and Gemini 2.5 Pro (Google, 2025). These models push the boundaries of multimodal reasoning, achieving strong performance in automatic speech recognition (ASR) (Hurst et al., 2024), cross-modal translation (Team et al., 2024a), and audio-visual captioning (Han et al., 2024; Zhan et al., 2024).

Benchmarking is a critical component of the community, as it helps specify the development direction. Prior work mainly focus on visual problem-solving, such as general comprehension (Li et al., 2024b; Liu et al., 2023c; Fu et al., 2023) and mathematical reasoning (Chen et al., 2021; Cao & Xiao, 2022; Chen et al., 2022; Zhang et al., 2025; Lu et al., 2023). On the other hand, audio-visual benchmarks such as AVQA (Yang et al., 2022), OmniBench (Li et al., 2025), and MusicAVQA (Li et al., 2022) focus on testing MLLMs with audio-visual tasks that require the simultaneous processing and integration of visual and auditory information.

In this paper, we directly test MLLMs'a ability to *see* instead of reasoning on simple low-level audio task. This is inspired by the BlindTest (Rahmanzadehgervi et al., 2024), which reveals that powerful

---

[1]In this work, MLLMs specifically refer to audio-vision LLMs, distinct from vision-only VLMs.

vision language models are still struggling with very simple vision tasks that are easy for humans. Concretely, we propose a DeafTest benchmark (Fig. 1), including four extremely simple audio tasks inspired from Schwabach test (Huizing, 1975). We test a set of powerful MLLMs like Gemini 1.5 (Team et al., 2024a), Gemini 2.5 Pro (Google, 2025), Reka (Team et al., 2024b), and GPT-4o (Hurst et al., 2024) on these easy task that only involve basic sound properties (*e.g.*, loudness, pitch, duration), as shown in Table 1. Our key findings are:

- Despite their ability to recognize complex speech content, MLLMs do not perform as well as expected on sound counting tasks. The best-performing model, Gemini 1.5 Pro, achieves only 81%, while humans can easily score 100%. The sounds in these tasks are monotonous and are clearly separated by silent intervals within the audio clip.

- Most MLLMs appear to be insensitive to sound loudness or sound pitch, except for Gemini 2.5 Pro and Qwen-2.5-Omni-7B. Models are required to distinguish the louder sound or the higher pitch from two given sounds. Several models perform under 60%, while a random guess baseline is 50%.

- The duration comparison task presents models with two sounds and asks them to determine which has the longer duration. Model performance varies significantly, where Gemini 2.5 Pro achieves a high score of 99.0%, models such as Reka Core and Reka Edge perform poorly, with scores of 40.0% and 44.0%, respectively.

These findings are noteworthy: despite the strong performance of proprietary models on complex tasks such as automatic speech recognition, their performance on basic audio perception tasks reveals a noticeable gap compared to human capabilities. Notably, Gemini 2.5 Pro (Google, 2025) generally outperforms other models. This raises a critical question: Does a deficiency in low-level audio perception limit a model's ability to perform higher-level audio-visual reasoning? Furthermore, can the performance on DeafTest serve as a key indicator of a model's overall audio-visual reasoning ability?

To address this, we introduce a novel and comprehensive benchmark **AV-Odyssey** to evaluate the audio-visual reasoning performance of MLLMs. This benchmark comprises 4,555 questions across 26 tasks, spanning a wide range of audio attributes, each strategically engineered to require cross-modal synergy (Fig. 3). Key design principles include: 1) multimodal necessity: questions are filtered using VLMs and audio models to exclude tasks solvable by single-modal reasoning; 2) diverse scopes: coverage of sound attributes (timbre, spatial dynamics), application domains (music, transportation), and temporal reasoning; 3) objective evaluation: multiple-choice format eliminates reliance on subjective human or LLM-based assessments.

We benchmark closed-source (GPT-4o (Hurst et al., 2024), Gemini 1.5 (Team et al., 2024a), Gemini 2.5 Pro (Google, 2025)) and open-source models (Han et al., 2024; Lu et al., 2022; Zhan et al., 2024; Wu et al., 2023; Su et al., 2023; Cheng et al., 2024; Fu et al., 2024) on our proposed AV-Odyssey. Combined with DeafTest results, the key findings are:

- Overall, current MLLMs still fall short in processing complex audio-visual information integration tasks.

- Performance on DeafTest strongly correlates with AV-Odyssey accuracy with a Pearson's $r = 0.945$ (Fig. 2). In other words, models that perform poorly on simple audio tasks (DeafTest) also consistently underperform on AV-Odyssey.

- Error analysis (see Sec. 4.3) shows that most audio-visual inference errors stem from misperceiving audio inputs. This finding aligns with findings on DeafTest, where models with weak performance on simple auditory tasks also underperform on AV-Odyssey.

To conclude, this work systematically investigates the audio-visual comprehension capabilities of current MLLMs through two complementary perspectives: DeafTest and AV-Odyssey Bench. **Our main contributions are as follows**: (1) DeafTest provides the **first** dedicated evaluation of MLLMs' basic listening abilities (*e.g.*, pitch and loudness discrimination), uncovering critical weaknesses in fundamental audio perception. (2) AV-Odyssey Bench introduces a large-scale and comprehensive benchmark for advanced cross-modal reasoning, spanning diverse audio attributes, visual contexts, and real-world scenarios. (3) We empirically demonstrate a strong correlation between fundamental audio perception and overall audio-visual reasoning performance, highlighting the dependency

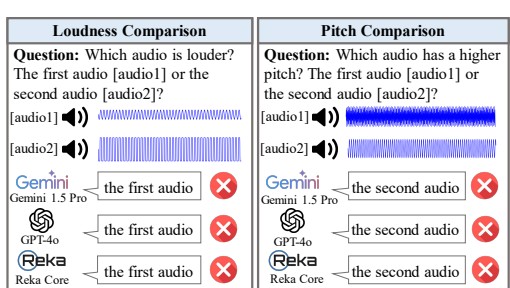

Figure 1: Illustration of two out of four Deaf-Test tasks: loudness and pitch comparison.

Figure 2: Performance on DeafTest and AV-Odyssey, showing a strong positive correlation.

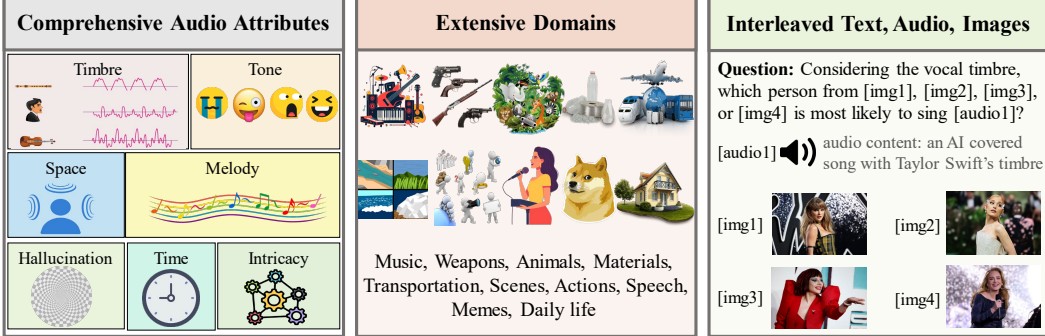

Figure 3: Overview of AV-Odyssey Benchmark. AV-Odyssey Bench demonstrates three major features: 1. Comprehensive Audio Attributes; 2. Extensive Domains; 3. Interleaved Text, Audio, and Images.

of higher-level multimodal reasoning on low-level perceptual skills. By explicitly linking low-level perception to high-level reasoning, our benchmarks offer the first comprehensive diagnosis of MLLMs' multimodal capabilities. We reveal that when pursuing strong audio-visual comprehension ability, it's noteworthy to monitor the model performance on fundamental audio perception performance. These resources provide a foundation for targeted advances in dataset construction, alignment strategies, and model architectures, ultimately paving the way toward truly integrated audio-visual intelligence.

## 2 RELATED WORK

**Multimodal Large Language Models**. Large language models (LLMs) have demonstrated remarkable performance across diverse textual domains (OpenAI, 2023; Radford, 2018; Brown, 2020; Touvron et al., 2023; Bi et al., 2024). The success of these models has catalyzed significant advancements in vision language models and multimodal large language models. Inspired by the textual prowess of LLMs, vision language models have emerged to extend computational capabilities into visual comprehension. These models enable LLMs to perform sophisticated visual tasks, including visual question answering (Liu et al., 2023a; Li et al., 2023a; Zhu et al., 2023; Liu et al., 2023b; Ye et al., 2023; Dai et al., 2023; Bai et al., 2023; Zhang et al., 2023b), visual grounding (Peng et al., 2023; Wang et al., 2023; Chen et al., 2023a;b), document understanding (Ye et al., 2023; Hu et al., 2024; Zhang et al., 2023c; Lv et al., 2023), long video understanding (Liu et al., 2024; Shen et al., 2024; Zhang et al., 2024a; Li et al., 2024c; Ren et al., 2024). Building upon vision-language achievements, researchers have further expanded multimodal horizons by integrating the audio modality (Han et al., 2024; Lu et al., 2022; Zhan et al., 2024; Wu et al., 2023; Su et al., 2023; Fu et al., 2024; Cheng et al., 2024; Chowdhury et al., 2024; Xu et al., 2025; Lu et al., 2025) . These advanced models now accommodate audio inputs, further expanding the landscape of multimodal artificial intelligence.

**Benchmarking Multimodal Large Language Models**. The rapid development of vision language models has been accompanied by the emergence of specialized benchmarks to assess their perfor-

Table 1: Results on four basic auditory tasks (DeafTest). The questions are designed as two-choice questions. The random baseline performance is 50%. The final column shows the average performance across the four tasks.

| Method | Sound Counting | Loudness Comparison | Pitch Comparison | Duration Comparison | Average |
|---|---|---|---|---|---|
| Random | 50.0 | 50.0 | 50.0 | 50.0 | 50.0 |
| Gemini 1.5 Flash (Team et al., 2024a) | 55.0 | 62.0 | 54.0 | 89.0 | 65.0 |
| Gemini 1.5 Flash-8B (Team et al., 2024a) | 49.0 | 55.0 | 51.0 | 51.0 | 51.5 |
| Gemini 1.5 Pro (Team et al., 2024a) | **81.0** | 60.0 | 52.0 | 84.0 | 69.3 |
| Gemini 2.5 Pro (Google, 2025) | 69.0 | 100.0 | **98.0** | 99.0 | **91.5** |
| Reka Core (Team et al., 2024b) | 54.0 | 43.0 | 42.0 | 40.0 | 44.8 |
| Reka Flash (Team et al., 2024b) | 48.0 | 58.0 | 51.0 | 44.0 | 50.3 |
| Reka Edge (Team et al., 2024b) | 47.0 | 56.0 | 50.0 | 44.0 | 49.3 |
| GPT-4o audio-preview (Hurst et al., 2024) | 50.0 | 58.0 | 58.0 | 57.0 | 55.8 |
| Qwen-2.5-Omni-7B (Xu et al., 2025) | 60.0 | 100.0 | 86.0 | 80.0 | 81.5 |

Table 2: Comparisons between MLLM benchmarks / datasets.

| Benchmark / Dataset | Modality | Questions | Answer Type | Customized Question | Audio Attributes | | | | | | | Multiple Domains | Interleaved |
|---|---|---|---|---|---|---|---|---|---|---|---|---|---|
| | | | | | Timbre | Tone | Melody | Space | Time | Hallucination | Intricacy | | |
| MME Bench (Fu et al., 2023) | Image | 2194 | Y/N | ✓ | - | - | - | - | - | - | - | ✓ | ✗ |
| MMBench (Liu et al., 2023c) | Image(s) | 2974 | A/B/C/D | ✓ | - | - | - | - | - | - | - | ✓ | ✗ |
| SEED-Bench-2 (Li et al., 2024b) | Image(s) & Video | 24371 | A/B/C/D | ✓ | - | - | - | - | - | - | - | ✓ | ✓ |
| AVQA Dataset (Yang et al., 2022) | Video & Audio | 57335 | A/B/C/D | ✓ | ✓ | ✗ | ✗ | ✗ | ✓ | ✗ | ✗ | ✓ | ✗ |
| Pano-AVQA Dataset (Yun et al., 2021) | Video & Audio | 51700 | defined words & bbox | ✓ | ✓ | ✓ | ✗ | ✓ | ✗ | ✗ | ✗ | ✓ | ✗ |
| Music-AVQA Dataset (Li et al., 2022) | Video & Audio | 45867 | defined words | ✓ | ✓ | ✗ | ✓ | ✓ | ✓ | ✓ | ✗ | ✓ | ✗ |
| SAVE Bench (Sun et al., 2024) | Image & Video & Audio | 4350 | free-form | ✗ | ✓ | ✗ | ✗ | ✓ | ✗ | ✗ | ✗ | ✓ | ✗ |
| OmniBench (Li et al., 2025) | Image & Audio | 1142 | A/B/C/D | ✓ | ✓ | ✗ | ✗ | ✗ | ✓ | ✗ | ✗ | ✓ | ✗ |
| AV-Odyssey Bench (ours) | Image(s) & Video & Audio(s) | 4555 | A/B/C/D | ✓ | ✓ | ✓ | ✓ | ✓ | ✓ | ✓ | ✓ | ✓ | ✓ |

mance across various domains (Yue et al., 2024; Fu et al., 2023; Li et al., 2024b; Chen et al., 2021; Lu et al., 2023). A significant subset of these benchmarks focuses on vision comprehension (Yue et al., 2024; Fu et al., 2023; Li et al., 2024b;a) and mathematical reasoning capabilities (Chen et al., 2021; Cao & Xiao, 2022; Lu et al., 2023; Zhang et al., 2025; Yue et al., 2024; Seo et al., 2015). However, current audio-visual benchmarks (Yang et al., 2022; Li et al., 2022; Yun et al., 2021; Sun et al., 2024; Li et al., 2025; Leng et al., 2024; Sung-Bin et al., 2024; Tang et al., 2024) face significant limitations in comprehensively assessing multimodal large language models (MLLMs). First, they predominantly focus on high-level visual tasks and neglect to explore the basic auditory perception limitations. Secondly, they do not comprehensively evaluate all attributes of the audio, comparison is detailed in Table 2. This paper begins by evaluating basic audio tasks to highlight shortcomings in auditory perception and introduces the AV-Odyssey benchmark, covering diverse audio attributes and domains. By leveraging both evaluations, we reveal key limitations of existing models and demonstrate a strong correlation between low-level audio perception and higher-level cross-modal reasoning. We emphasize that effective audio-visual comprehension requires attention to fundamental audio perception, a factor not emphasized by previous work.

## 3 METHOD

### 3.1 DEAFTEST TASKS

Drawing inspiration from the Schwabach test (Huizing, 1975), we introduce DeafTest, a suite of four extremely simple auditory tasks that critically examine the fundamental audio perception capabilities of Multi-modal Large Language Models (MLLMs). DeafTest includes the determination of the number of sounds, identification of the louder sound, recognition of the sound with a higher pitch, and detection of the sound with a longer duration. We hypothesize that MLLMs may not perform as well as expected on these basic tasks. This potential shortcoming arises from the training objectives of these models, which primarily focus on achieving high-level semantic alignment between different modalities. Consequently, this approach tends to overlook the effective utilization of low-level auditory information, which is crucial for accurately processing and understanding basic sound characteristics.

**1. Count the Number of Sounds**. This task evaluates MLLMs' ability to count distinct sounds in audio clips containing 3 to 8 monotonous, clearly separated sounds. Despite strong ASR performance (*e.g.*, GPT-4o's 3% word error rate (OpenAI)), this tests basic auditory segmentation. Each trial presents a two-choice question. We curate 100 questions in total.

Table 3: Detailed statistics of the AV-Odyssey Benchmark.

| Statistics | Number |
| --- | --- |
| Total Questions | 4555 |
| Total Tasks | 26 |
| Domains | 10 |
| Multi-Image, Single-Audio | 2610 |
| Single-Image, Multi-Audio | 891 |
| Single-Image, Single-Audio | 434 |
| Single-Video, Single-Audio | 220 |
| Single-Video, Multi-Audio | 400 |
| Correct Option Dist. (A:B:C:D) | 1167:1153:1119:1116 |
| Avg. Audio Time (s) | 16.32 |
| Avg. Image Res. (px) | 1267×891 |
| Avg. Video Res. (px) | 1678×948 |
| Avg. Video Time (s) | 15.58 |

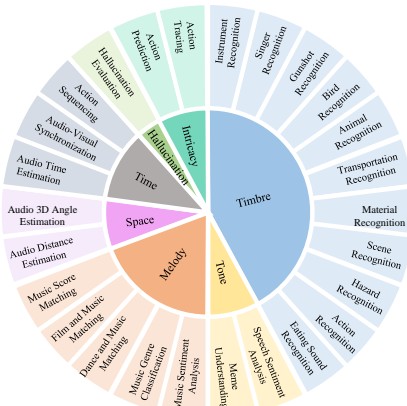

Figure 4: Overview of the 26 evaluation tasks, categorized into 7 classes based on sound attributes.

**2. Discriminate the Louder Sound**. In this task, we test the basic ability of MLLMs to distinguish between the loudness of sounds. The goal of MLLMs is to discriminate which sound is louder out of two given audio clips. Specifically, the decibel for quieter audio ranges from 30 dB to 60 dB, while the decibel for louder audio ranges from 70 dB to 100 dB. We randomly sample decibels from these two ranges to create two audio clips. In addition, we randomly switch the input order of the two audio clips; that is, for some questions, the quiet audio comes first, and for the rest, the loud audio comes first. Similarly, the question format is also a two-choice question.

**3. Discriminate the Higher Pitch**. This task focuses on pitch differentiation. Clips feature lower-pitched sounds (100–500 Hz) vs. higher-pitched ones (1000–2000 Hz), validated for human discernibility. Input order is randomized, with two-choice questions (100 total).

**4. Recognize the Duration of Sound**. We also test MLLMs with the duration of sound. In this task, we simplify the question by giving two audio clips of different durations. We sample the duration from 1s to 3s for the short audio, while we sample from 4s to 6s for the long audio. Similar to task 2, we provide the MLLMs with two audio clips, asking them to identify the longer one.

The DeafTest results in Table 1 reveal significant variation in model performance, with Gemini 2.5 Pro and Qwen-2.5-Omni-7B outperforming others. However, for all other models, performance falls far below expectations, with none exceeding 62%, particularly on tasks such as loudness and pitch comparison. These findings highlight substantial limitations in basic auditory perception among current MLLMs. This raises the question: could deficiencies in basic audio perception impair performance on higher-level audio-visual reasoning tasks? To investigate, we introduce the AV-Odyssey benchmark, which reveals a strong correlation between fundamental audio perception and overall audio-visual reasoning, as shown in Fig. 2.

## 3.2 OVERVIEW OF AV-ODYSSEY BENCH

Our AV-Odyssey Bench is a meticulously curated benchmark designed to comprehensively assess the audio-visual capabilities of MLLMs. To ensure a robust and unbiased assessment, all questions in AV-Odyssey are structured as multiple-choice, with four options per question, and options can be presented in various formats, including text, images, or audio clips. To mitigate format-specific biases, we have curated five distinct multi-choice question types. Additionally, all inputs, including text, image/video, and audio clips, are fed into MLLMs in an interleaved manner.

We compare our AV-Odyssey benchmark with previous MLLM benchmarks and datasets in Table 2. It can be found that previous works suffer from certain limitations, such as restricted audio attributes, which fail to capture the full spectrum of auditory complexity and the absence of interleaved settings, crucial for assessing real-world multimodal integration capabilities. For instance, OmniBench (Li et al., 2025) lacks multiple audio attributes, making it difficult to comprehensively assess the capabilities of MLLMs in audio-visual tasks. In contrast, our AV-Odyssey encompasses 26 tasks across 10 diverse domains and spans 7 audio attributes, with interleaved and customized

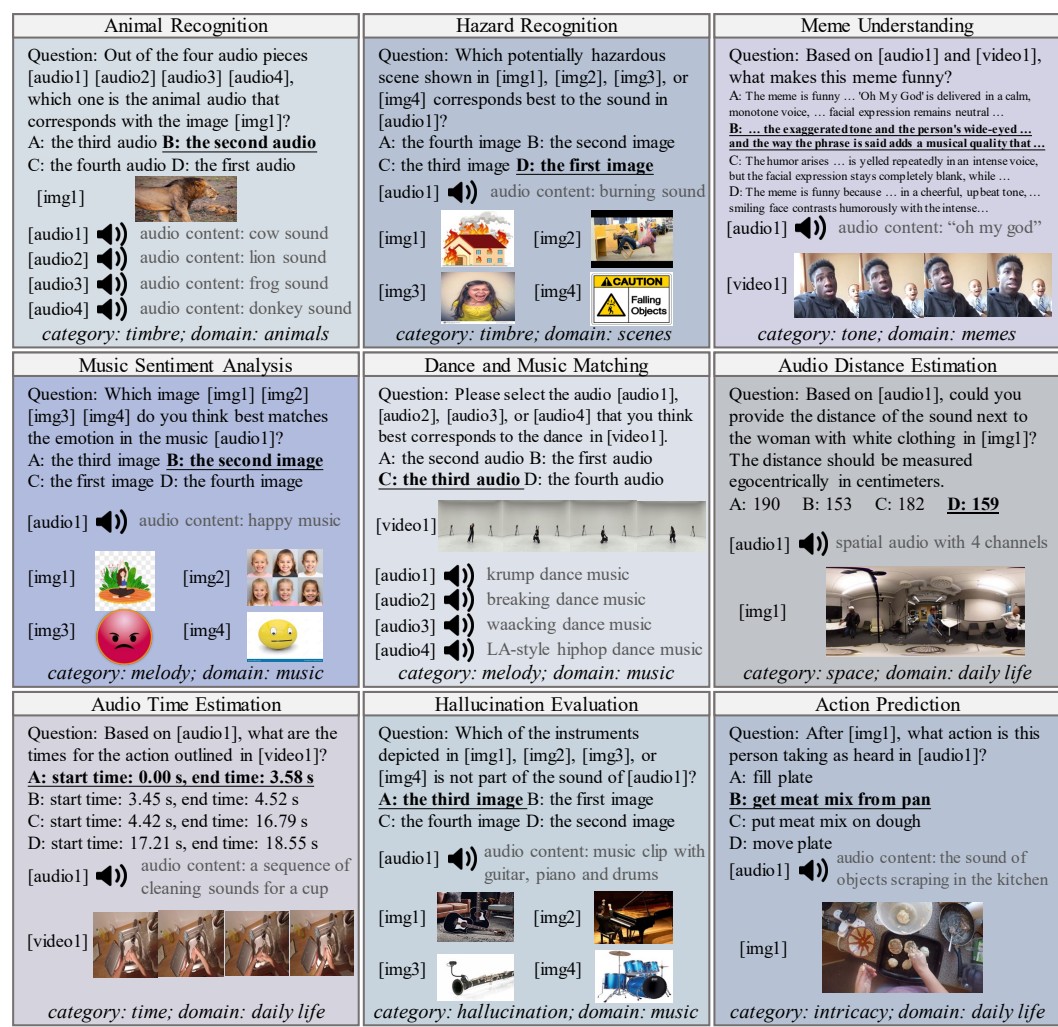

Figure 5: Sampled examples from our AV-Odyssey Benchmark.

questions. The detailed statistics are shown in Table 3. This design enables an exhaustive evaluation of MLLMs, providing a nuanced and thorough assessment of their performance in complex, real-world audio-visual scenarios.

Here, we will briefly introduce the task categories that span a broad spectrum of audio attributes, including Timbre, Tone, Melody, Spatial characteristics, Temporal dynamics, and Hallucination detection. The detailed task distribution and task examples are shown in Fig. 4 and Fig. 5, respectively.

**Timbre Tasks.** To test the concept of matching across vision and audio modalities, MLLMs are required to match audio-visual pairs (*e.g.*, lion's roar sound with lion images) in timbre tasks. In addition, we have designed advanced tasks that demand internal expert-level knowledge learned from the large-scale pretraining data to solve, such as singer recognition and bird species identification.

**Tone Tasks.** These tasks target evaluating MLLMs with speech sentiment analysis and meme understanding. For example, meme understanding requires MLLMs to infer humorous reasons simultaneously from the voice tone and visual context.

**Melody Tasks.** For evaluating melody understanding abilities, we propose melody tasks. For example, the dance and music matching task requires the MLLM to understand the melody of the music and identify the one that aligns with the dance in a video.

**Space Tasks.** To test the spatial inference with audio and visual information, space tasks require MLLMs to infer the distance of a certain object producing a sound or to determine the 3D angle.

**Time Tasks.** These tasks test the cross-modal matching and temporal correlation abilities at the same time. For example, audio time estimation requires MLLMs to determine the start and end time of an action.

**Hallucination Tasks**. Inspired by POPE (Li et al., 2023b) that indicates severe object hallucination existing in vision language models, we designed this task to assess the hallucination issue in audio-visual reasoning.

**Intricacy Tasks.** These tasks challenge MLLMs to perform integrated analysis or reasoning through both visual and audio inputs, leveraging multiple attributes. For example, action prediction requires models to infer actions based on visual elements alongside various audio attributes, such as timbre and timing.

These diverse tasks provide a rigorous and multifaceted assessment of MLLMs' audio-visual information integration capabilities, systematically probing the depth, nuance, and complexity of cross-modal perception and reasoning.

### 3.3 DATA CURATION PROCESS

**Data Collection**. AV-Odyssey Bench is an audio-visual benchmark to evaluate whether MLLMs truly have audio-visual reasoning capability. Since the audio is the newly added modality by these omni-modal models, and there is already an array of visual benchmarks, we put our attention on the attributes of sound in the benchmark construction. We first design 26 tasks that cover a wide range of audio attributes and application domains, then we manually curate each question.

**Data source**. Our questions involve audio, images, and videos. We use public datasets as an important source of fetching audios (MISC; AYDEMİR; Microsoft). For example, we use the instrument audios from MISC and bird sound audios from Microsoft. We mainly use images from the internet and manually filter out low-quality images. The sources of videos are either from existing datasets (Shimada et al., 2024; Damen et al., 2018) or from the internet. For example, we download meme videos from video websites to create a meme understanding question. We will detail the data source of each task in the appendix.

**Quality Control**. Each question-answer pair is verified by two different experts. Duplicated text information that describes visual inputs will induce MLLMs to bypass the visual input to directly derive the answer by memorizing the answer from the internet-scale training dataset (Chen et al., 2024a). Inspired by this, we first ensure that our text questions' context is as simple as possible. Then we filter out those questions that have redundant images or audio clips by leveraging VLMs and audio LLMs. Specifically, we test all the curated questions with VLM: InternVL2 (Chen et al., 2024b), Qwen2-VL (Wang et al., 2024), MiniCPM-V 2.5 (Yao et al., 2024), BLIP3 (Xue et al., 2024), and VILA1.5 (Lin et al., 2024) and audio LLM Qwen-Audio (Chu et al., 2023), Qwen2-Audio (Chu et al., 2024), SALMONN (Tang et al., 2023), and Typhoon-Audio (Manakul et al., 2024), and filter out those questions that either of these models can solve. In experiment, 2.54% questions are filtered out because they are solved by all audio LLMs or VLMs

## 4 EXPERIMENT

**Model Testing.** We evaluate closed-source and open-source MLLMs supporting text, image/video, and audio inputs. Experiments are conducted in a **zero-shot setting** (no finetuning or few-shot prompting) to assess inherent multimodal reasoning capabilities. **Prompt Design.** Text prompts are minimized to exclude redundant information. To ensure robustness: 1. Predefine multiple question templates, randomly selecting one per evaluation. 2. Conduct three trials per question to mitigate stochasticity in model outputs.

### 4.1 MODEL EVALUATION AND METHODOLOGY

**Evaluated Models.** We assess 21 models: 9 closed-source (Gemini 2.5 Pro (Google, 2025), Gemini 1.5 Flash/Pro (Team et al., 2024a), Reka Core/Flash/Edge (Team et al., 2024b), GPT-4o (Hurst et al., 2024)) and 12 open-source (Qwen-2.5-Omni-7B (Xu et al., 2025), AV-Reasoner (Lu et al., 2025), Unified-IO-2 L/XL/XXL (Lu et al., 2022), PandaGPT (Su et al., 2023), VideoLLaMA (Zhang et al.,

Table 4: Evaluation results of various MLLMs in different parts of AV-Odyssey Bench. The highest performance is highlighted in bold. $\bar{T}$ is the averaged accuracy across corresponding dimensions, and $R_{\bar{T}}$ is the rank based on the averaged accuracy. "All Avg." represents the averaged accuracy over all questions in our AV-Odyssey Bench.

| | Model | LLM Size | Timbre | | Tone | | Melody | | Space | | Time | | Hallucination | | Intricacy | | All Avg. | |
|---|---|---|---|---|---|---|---|---|---|---|---|---|---|---|---|---|---|---|
| | | | $\bar{T}$ | $R_{\bar{T}}$ | $\bar{T}$ | $R_{\bar{T}}$ | $\bar{T}$ | $R_{\bar{T}}$ | $\bar{T}$ | $R_{\bar{T}}$ | $\bar{T}$ | $R_{\bar{T}}$ | $\bar{T}$ | $R_{\bar{T}}$ | $\bar{T}$ | $R_{\bar{T}}$ | $\bar{T}$ | $R_{\bar{T}}$ |
| | Random | - | 25.0 | 14 | 25.0 | 11 | 25.0 | 20 | 25.0 | 16 | 25.0 | 19 | 25.0 | 14 | 25.0 | 18 | 25.0 | 21 |
| Open Source | Unified-IO-2 L (Lu et al., 2022) | 1B | 23.8 | 20 | 24.1 | 14 | 28.8 | 8 | 15.0 | 22 | 26.8 | 10 | 30.0 | 7 | 30.4 | 12 | 26.0 | 18 |
| | Unified-IO-2 XL (Lu et al., 2022) | 3B | 24.3 | 17 | 23.2 | 16 | 27.8 | 9 | 22.5 | 19 | 25.3 | 18 | 31.5 | 4 | 34.8 | 5 | 26.3 | 15 |
| | Unified-IO-2 XXL (Lu et al., 2022) | 7B | 26.3 | 9 | 22.7 | 18 | 26.4 | 14 | 32.5 | 6 | 26.8 | 10 | 24.5 | 16 | 33.8 | 8 | 27.2 | 9 |
| | OneLLM (Han et al., 2024) | 7B | 25.0 | 15 | 25.5 | 9 | 21.5 | 22 | 37.5 | 5 | **29.3** | 1 | 25.5 | 13 | 38.4 | 3 | 27.4 | 8 |
| | PandaGPT (Su et al., 2023) | 7B | 23.5 | 19 | 23.2 | 16 | 27.6 | 12 | **45.0** | 1 | 23.8 | 21 | 28.0 | 12 | 23.9 | 20 | 26.7 | 13 |
| | Video-llama (Zhang et al., 2023a) | 7B | 25.5 | 10 | 22.3 | 19 | 24.4 | 21 | 30.0 | 9 | 26.2 | 15 | 25.0 | 14 | 30.7 | 11 | 26.1 | 17 |
| | VideoLLaMA2 (Cheng et al., 2024) | 7B | 24.1 | 18 | 25.5 | 9 | 26.4 | 16 | 30.0 | 9 | 27.2 | 9 | 33.0 | 2 | 34.5 | 6 | 26.8 | 12 |
| | AnyGPT (Zhan et al., 2024) | 7B | 24.6 | 16 | 25.0 | 11 | 26.4 | 17 | 27.5 | 14 | 29.2 | 2 | 29.0 | 8 | 25.7 | 17 | 26.1 | 19 |
| | NExT-GPT (Wu et al., 2023) | 7B | 23.2 | 22 | 20.9 | 20 | 27.8 | 11 | 30.0 | 9 | 28.8 | 3 | 28.5 | 10 | 23.6 | 21 | 25.5 | 20 |
| | VITA (Fu et al., 2024) | 8 × 7B | 24.1 | 19 | 26.4 | 8 | 27.8 | 9 | 22.5 | 19 | 26.3 | 14 | 31.0 | 6 | 36.8 | 4 | 26.4 | 14 |
| | Qwen-2.5-Omni-7B (Xu et al., 2025) | 7B | 38.6 | 4 | 30.0 | 6 | 30.4 | 7 | 40.0 | 3 | 25.8 | 16 | 31.5 | 4 | 39.6 | 2 | 34.5 | 4 |
| | AV-Reasoner (Lu et al., 2025) | 7B | 42.5 | 2 | 31.4 | 4 | 28.3 | 9 | 40.0 | 3 | 26.5 | 12 | 32.5 | 3 | **43.4** | 1 | 36.4 | 2 |
| Closed Source | Gemini 1.5 Flash (Team et al., 2024a) | - | 27.2 | 7 | 25.0 | 11 | 28.8 | 8 | 30.0 | 9 | 25.3 | 18 | 28.5 | 10 | 31.2 | 10 | 27.8 | 7 |
| | Gemini 1.5 Flash-8B (Team et al., 2024a) | - | 25.1 | 13 | 24.5 | 13 | 28.9 | 6 | 27.5 | 14 | 27.5 | 5 | 29.0 | 5 | 30.2 | 13 | 26.8 | 13 |
| | Gemini 1.5 Pro (Team et al., 2024a) | - | 30.8 | 6 | 31.4 | 4 | 31.3 | 5 | 37.5 | 5 | 27.7 | 4 | 20.5 | 21 | 33.0 | 9 | 30.8 | 6 |
| | Gemini 2.5 Pro (Google, 2025) | - | **53.6** | 1 | **40.0** | 1 | **41.9** | 1 | 32.5 | 6 | 23.8 | 21 | **40.5** | 1 | 44.4 | 1 | **44.4** | 1 |
| | Reka Core (Team et al., 2024b) | 67B | 26.7 | 8 | 27.7 | 7 | 26.4 | 15 | 22.5 | 19 | 26.5 | 12 | 24.0 | 17 | 34.3 | 7 | 26.9 | 10 |
| | Reka Flash (Team et al., 2024b) | 21B | 25.5 | 11 | 24.1 | 14 | 27.2 | 13 | 30.0 | 9 | 27.5 | 5 | 31.5 | 4 | 24.1 | 19 | 26.3 | 16 |
| | Reka Edge (Team et al., 2024b) | 7B | 23.8 | 21 | 20.5 | 22 | 26.3 | 18 | 22.5 | 19 | 25.5 | 17 | 22.5 | 19 | 36.8 | 4 | 25.0 | 22 |
| | GPT-4o visual caption (Hurst et al., 2024) | - | 37.4 | 5 | 28.6 | 6 | 32.3 | 4 | 27.5 | 14 | 25.5 | 17 | 23.0 | 18 | 28.9 | 14 | 32.3 | 5 |
| | GPT-4o audio caption (Hurst et al., 2024) | - | 38.6 | 3 | 31.8 | 2 | 33.6 | 2 | 42.5 | 2 | 27.5 | 5 | 25.0 | 15 | 26.1 | 16 | 34.5 | 3 |

2023a), OneLLM (Han et al., 2024), AnyGPT (Zhan et al., 2024), NExT-GPT (Wu et al., 2023), VITA (Fu et al., 2024)). Open-source models were tested using their latest checkpoints and **default hyperparameters** from published code; closed-source models relied on official APIs.

**GPT-4o Workaround.** Due to API limitations preventing simultaneous multimodal inputs, we evaluate GPT-4o via two pipelines: 1. *Audio Caption Method*: We use GPT-4o-audio to generate audio captions, then use the audio caption, question text, and visual as the inputs of GPT-4o. 2. *Visual Caption Method*: We use GPT-4o to generate visual captions, then use the visual caption, question text, and audio as the inputs of GPT-4o-audio.

**Baseline and Interpretation.** A 25% random baseline (corresponding to a four-choice task) is established. Performance below this threshold indicates the model's inability to tackle the task.

## 4.2 Experimental Analysis

In this section, we analyze the performance of MLLMs in our AV-Odyssey benchmark, as presented in Table 4. We showcase the mean accuracy of each audio attribute. Detailed results and data distribution are provided in the Appendix. Our key findings are as follows:

**1. Challenging Nature of AV-Odyssey.** Table 4 shows that most MLLMs achieve barely above 25% accuracy—only marginally better than random guessing. The top-performing model, Gemini 2.5 Pro, reaches just 44.4%. These results highlight the rigor of AV-Odyssey, which evaluates capabilities beyond the scope of current training data. By establishing demanding standards, AV-Odyssey serves as a crucial benchmark for assessing MLLM robustness and versatility in audio-visual reasoning, revealing key limitations and providing insights for future advancements.

**2. Comparison Between Audio Captions and Visual Captions.** It can be observed that GPT-4o audio caption achieves higher performance than GPT-4o visual caption as shown in Table 4. However, drawing conclusions from this comparison is challenging, as the captions are generated by the model itself, introducing potential biases. To isolate the influence of each modality, we further investigate this phenomenon by providing ground-truth (GT) captions for both audio and visual tasks. In this setup, GPT-4o's performance improves to 73.33% with ground-truth audio

| | w/ GT audio caption | w/ GT visual caption |
|---|---|---|
| GPT-4o | 73.33% | 47.33% |

Table 5: GPT-4o with GT audio/visual caption on randomly selected 300 Questions.

captions, while performance with ground-truth visual captions increases to 47.33%. These results suggest that the primary limitation of current models lies in their audio understanding capabilities.

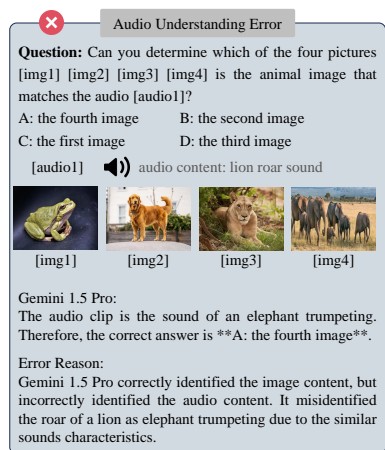

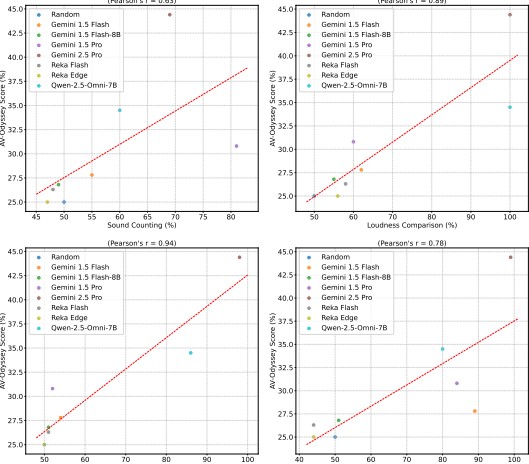

Figure 6: An example of audio understanding error. More examples are provided in the Appendix.

Figure 7: Comparison performance between each Deaf-Test task and AV-Odyssey.

**3. Correlation between each DeafTest task and AV-Odyssey**. We analyze the relationship between performance on DeafTest and AV-Odyssey (Fig. 7). Notably, loudness and pitch perception show a stronger correlation with AV-Odyssey performance, as these fundamental auditory components are crucial for accurate audio understanding. Tasks such as audio length recognition and sound counting exhibit weaker correlations with AV-Odyssey performance. This is because AV-Odyssey is designed to evaluate comprehensive audio-visual understanding. As such, tasks focused on isolated auditory features, such as temporal duration or discrete event counting, are less strongly aligned with the broader, more integrated tasks within AV-Odyssey.

### 4.3 Error Analysis

We analyzed 104 human-annotated errors (4 randomly sampled per task) to identify failure modes. The error distribution is shown in Fig. 8, with case details in the Appendix.

**1. Perception Understanding Errors (81%).** Most errors stemmed from flawed input interpretation, **dominated by audio-related failures (63%)**. For example, Fig. 6 shows a misidentified audio clip leading to incorrect answers. Vision (10%) and text (8%) understanding errors were less frequent. This **aligns with DeafTest findings**: deficient audio perception undermines cross-modal integration. **2. Reasoning Errors (13%).** In these cases, Gemini 1.5 Pro correctly parsed audio/visual inputs but failed in logical inference (*e.g.*, misconnecting cause-effect relationships). **3. Other Errors (6%).** Mostly rejected responses (*e.g.*, content flagged for security).

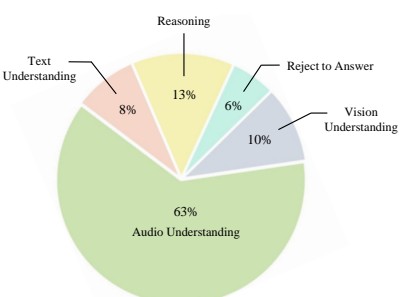

Figure 8: Distribution of 104 human-annotated errors in the Gemini 1.5 Pro.

## 5 Limitations and Future Work

In this work, we introduce DeafTest and AV-Odyssey to evaluate MLLMs from fundamental audio perception and audio-visual reasoning perspectives. Further, we reveal the strong correlation between basic audio perception and complex audio-visual reasoning.

Our study does not yet address long video understanding (*e.g.*, five minutes or more), real-time interactive scenarios, or joint understanding–generation tasks. These directions remain promising avenues for future work. While our results underscore the importance of fundamental audio perception, additional factors influencing audio-visual reasoning are likely to exist and warrant further investigation.

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

# A   APPENDIX

# B   USE OF LLM

We employed large language models (LLMs), such as ChatGPT, to assist in manuscript preparation. Their role was restricted to language refinement (including grammar, spelling, and word choice), code formatting (e.g., adding explanatory comments), and drafting preliminary figures to support the development of final visualizations. All scientific concepts, analyses, and conclusions were independently conceived, verified, and interpreted by the authors. We sincerely acknowledge the valuable support of LLMs in this work.

# C   DATA DISTRIBUTION

In this section, we present the detailed data distribution of our AV-Odyssey Bench in Table 6. Our AV-Odyssey bench consists of 26 tasks covering a wide range of task categories.

**We will make all the data and evaluation codes public.**

Table 6: Detailed task statistics in AV-Odyssey Bench.

| Task ID | Task Name | Task Category | Class | Number | Data Source |
|---|---|---|---|---|---|
| 1 | Instrument Recognition | Timbre | 28 | 200 | MISC |
| 2 | Singer Recognition | Timbre | 20 | 200 | Internet |
| 3 | Gunshot Recognition | Timbre | 13 | 200 | AYDEMİR |
| 4 | Bird Recognition | Timbre | 39 | 200 | Microsoft |
| 5 | Animal Recognition | Timbre | 13 | 200 | Putthewad |
| 6 | Transportation Recognition | Timbre | 8 | 200 | AKBAL et al. (2021) |
| 7 | Material Recognition | Timbre | 10 | 200 | Sterling et al. (2018) |
| 8 | Scene Recognition | Timbre | 8 | 200 | Heittola et al. (2019) |
| 9 | Hazard Recognition | Timbre | 8 | 108 | Kay et al. (2017) |
| 10 | Action Recognition | Timbre | 20 | 196 | Soomro (2012) |
| 11 | Eating Sound Recognition | Timbre | 20 | 200 | Ma |
| 12 | Speech Sentiment Analysis | Tone | 7 | 200 | Lok |
| 13 | Meme Understanding | Tone | N/A | 20 | Internet |
| 14 | Music Sentiment Analysis | Melody | 7 | 197 | yash bhaskar |
| 15 | Music Genre Classification | Melody | 8 | 200 | Andrada |
| 16 | Dance and Music Matching | Melody | 10 | 200 | Tsuchida et al. (2019) |
| 17 | Film and Music Matching | Melody | 5 | 200 | Defferrard et al. (2016) |
| 18 | Music Score Matching | Melody | N/A | 200 | Zhang et al. (2024b) |
| 19 | Audio 3D Angle Estimation | Space | N/A | 20 | Shimada et al. (2024) |
| 20 | Audio Distance Estimation | Space | N/A | 20 | Shimada et al. (2024) |
| 21 | Audio Time Estimation | Time | N/A | 200 | Damen et al. (2018) |
| 22 | Audio-Visual Synchronization | Time | N/A | 200 | Tian et al. (2018) |
| 23 | Action Sequencing | Time | N/A | 200 | Damen et al. (2018) |
| 24 | Hallucination Evaluation | Hallucination | 19 | 200 | Ostermann et al. (2023) |
| 25 | Action Prediction | Intricacy | N/A | 199 | Damen et al. (2018) |
| 26 | Action Tracing | Intricacy | N/A | 195 | Damen et al. (2018) |

# D    BREAKDOWN RESULTS

In this section, we provide detailed results of evaluated methods on our proposed AV-Odyssey Bench, as demonstrated in Table 7 and Table 8.

Table 7: Evaluation results of various MLLMs in 'Timbre' part of AV-Odyssey Bench. The best performance is in bold. The corresponding brackets for each task indicate the number of associated questions.

| | Model | LLM Size | Instrument Recognition | Singer Recognition | Gunshot Recognition | Bird Recognition | Animal Recognition | Transportation Recognition | Material Recognition | Scene Recognition | Hazard Recognition | Action Recognition | Eating Sound Recognition |
|---|---|---|---|---|---|---|---|---|---|---|---|---|---|
| | | | (200) | (200) | (200) | (200) | (200) | (200) | (200) | (200) | (108) | (196) | (200) |
| Open Source | Unified-IO-2 L (Lu et al., 2022) | 1B | 20.5 | 22.5 | 25.5 | 18.5 | 27.0 | 26.5 | 23.0 | 28.0 | 21.3 | 20.9 | 26.5 |
| | Unified-IO-2 XL (Lu et al., 2022) | 3B | 20.0 | 23.5 | 24.0 | 20.5 | 27.5 | 26.0 | 27.5 | 30.0 | 19.4 | 19.9 | 26.5 |
| | Unified-IO-2 XXL (Lu et al., 2022) | 7B | 29.5 | 24.0 | 23.5 | **29.0** | 23.5 | 25.5 | 30.5 | 26.5 | 23.1 | 27.0 | 25.5 |
| | OneLLM (Han et al., 2024) | 7B | 26.0 | 21.5 | 27.0 | 26.0 | 22.0 | 20.0 | 29.5 | 24.5 | 26.9 | 23.0 | 29.5 |
| | PandaGPT (Su et al., 2023) | 7B | 20.0 | 21.5 | 23.0 | 17.5 | 26.0 | 26.5 | 28.0 | 27.0 | 23.1 | 21.4 | 24.5 |
| | Video-llama (Zhang et al., 2023a) | 7B | 22.5 | 24.5 | 27.0 | 26.5 | 27.0 | 23.5 | 28.0 | 25.0 | 25.0 | 26.0 | 25.5 |
| | VideoLLaMA2 (Cheng et al., 2024) | 7B | 22.5 | 24.0 | 27.0 | 17.0 | 23.5 | 27.5 | 26.5 | 26.5 | 19.4 | 23.0 | 25.5 |
| | AnyGPT (Zhan et al., 2024) | 7B | 22.5 | 28.5 | 28.0 | 17.5 | 24.0 | 25.5 | 23.0 | 28.0 | 25.9 | 20.4 | 27.5 |
| | NExT-GPT (Wu et al., 2023) | 7B | 21.0 | 23.5 | 25.5 | 21.5 | 25.5 | 25.5 | 21.0 | 24.0 | 19.4 | 23.0 | 24.0 |
| | VITA (Fu et al., 2024) | 8 × 7B | 22.0 | 20.5 | 24.5 | 21.5 | 27.5 | 25.0 | 23.5 | 28.5 | 21.3 | 19.4 | 29.5 |
| | Qwen-2.5-Omni-7B (Xu et al., 2025) | 7B | 52.0 | 26.0 | 24.0 | 28.5 | 66.0 | 34.0 | 32.5 | 33.5 | 45.4 | 61.7 | 23.0 |
| | AV-Reasoner (Lu et al., 2025) | 7B | 56.5 | 28.5 | 32.0 | 33.5 | 53.5 | 48.5 | 36.5 | 37.5 | 47.2 | 67.9 | 27.0 |
| Closed Source | Gemini 1.5 Flash (Team et al., 2024a) | - | 24.5 | 24.0 | 23.5 | 17.0 | 32.5 | 26.0 | 22.5 | 29.5 | 34.3 | 48.0 | 21.5 |
| | Gemini 1.5 Flash-8B (Team et al., 2024a) | - | 16.5 | 22.5 | 24.0 | 19.0 | 28.0 | 26.5 | 27.0 | 29.0 | 26.9 | 32.7 | 24.5 |
| | Gemini 1.5 Pro (Team et al., 2024a) | - | 33.0 | 26.0 | 29.0 | 25.0 | 25.5 | 26.0 | 29.5 | 30.0 | 38.0 | 57.7 | 22.5 |
| | Gemini 2.5 Pro (Google, 2025) | - | **79.0** | **41.0** | **39.5** | 21.0 | **76.0** | **60.5** | **39.0** | **44.0** | **65.7** | **87.8** | **42.5** |
| | Reka Core (Team et al., 2024b) | 67B | 32.5 | 20.0 | 26.5 | 25.0 | 24.0 | 27.0 | 30.0 | 27.0 | 25.0 | 34.2 | 21.5 |
| | Reka Flash (Team et al., 2024b) | 21B | 20.0 | 22.5 | 26.5 | 26.0 | 28.5 | 26.5 | 26.5 | 29.0 | 28.7 | 22.4 | 25.0 |
| | Reka Edge (Team et al., 2024b) | 7B | 21.5 | 24.0 | 30.5 | 20.0 | 19.5 | 22.5 | 20.5 | 25.5 | 25.9 | 23.5 | 29.0 |
| | GPT-4o visual caption (Hurst et al., 2024) | - | 33.0 | 30.5 | 24.0 | 26.5 | 43.0 | 42.0 | 32.5 | 39.0 | 49.1 | 67.3 | 30.5 |
| | GPT-4o audio caption (Hurst et al., 2024) | - | 40.0 | 38.0 | 27.5 | 26.5 | 45.0 | 42.0 | 27.0 | 41.0 | 42.6 | 62.2 | 35.5 |

Table 8: Evaluation results of various MLLMs in 'Time', 'Melody', 'Space'. 'Time', 'Hallucination', and 'Intricacy' parts of AV-Odyssey Bench. The best (second best) is in bold (underline). The corresponding brackets for each task indicate the number of associated questions.

| | Model | LLM Size | Tone | | Melody | | | | | Space | | Time | | | Hallucination | Intricacy | |
|---|---|---|---|---|---|---|---|---|---|---|---|---|---|---|---|---|---|
| | | | Speech Sentiment Analysis | Meme Understanding | Music Sentiment Analysis | Music Genre Classification | Dance and Music Matching | Film and Music Matching | Music Score Matching | Audio 3D Angle Estimation | Audio Distance Estimation | Audio Time Estimation | Audio-Visual Synchronization | Action Sequencing | Hallucination Evaluation | Action Prediction | Action Tracing |
| | | | (200) | (20) | (97) | (200) | (200) | (200) | (200) | (20) | (20) | (200) | (200) | (200) | (200) | (199) | (195) |
| Open Source | Unified-IO-2 L (Lu et al., 2022) | 1B | 24.5 | 20.0 | 27.9 | 31.0 | 27.5 | 32.5 | 24.5 | 15.0 | 15.0 | 28.0 | 25.5 | 27.0 | 30.0 | 27.1 | 33.8 |
| | Unified-IO-2 XL (Lu et al., 2022) | 3B | 23.0 | 25.0 | 26.9 | 30.5 | 27.0 | 31.5 | 22.5 | 30.0 | 15.0 | 26.5 | 25.5 | 24.0 | 31.5 | 35.7 | 33.8 |
| | Unified-IO-2 XXL (Lu et al., 2022) | 7B | 23.0 | 20.0 | 23.9 | 31.5 | 27.5 | 24.5 | 23.5 | 50.0 | 15.0 | 28.0 | 25.0 | 27.5 | 24.5 | 33.2 | 34.4 |
| | OneLLM (Han et al., 2024) | 7B | 26.0 | 20.0 | 20.8 | 23.5 | 26.5 | 18.5 | 18.0 | 45.0 | 30.0 | 31.5 | 29.5 | 27.0 | 25.5 | 41.7 | 34.9 |
| | PandaGPT (Su et al., 2023) | 7B | 23.5 | 20.0 | 21.6 | 28.0 | 27.0 | 32.5 | 26.0 | 45.0 | 45.0 | 18.5 | 26.0 | 27.0 | 28.0 | 19.6 | 28.2 |
| | Video-llama (Zhang et al., 2023a) | 7B | 23.0 | 15.0 | 25.8 | 24.0 | 20.0 | 25.0 | 28.0 | 45.0 | 15.0 | 28.5 | 23.5 | 26.5 | 25.0 | 28.6 | 32.8 |
| | VideoLLaMA2 (Cheng et al., 2024) | 7B | 26.0 | 20.0 | 26.8 | 29.0 | 25.5 | 30.5 | 20.5 | 45.0 | 15.0 | 28.5 | 26.5 | 26.5 | 33.0 | 28.6 | 40.5 |
| | AnyGPT (Zhan et al., 2024) | 7B | 25.5 | 20.0 | 23.4 | 29.5 | 25.5 | 26.0 | 26.0 | 40.0 | 15.0 | 30.5 | 28.0 | 29.0 | 29.0 | 21.1 | 30.3 |
| | NExT-GPT (Wu et al., 2023) | 7B | 21.5 | 15.0 | 23.7 | 26.0 | 28.0 | 31.0 | 28.0 | 45.0 | 15.0 | 31.5 | 24.0 | 31.0 | 28.5 | 20.6 | 26.7 |
| | VITA (Fu et al., 2024) | 8 × 7B | 24.5 | 45.0 | 26.8 | 26.0 | 27.5 | 33.5 | 24.5 | 25.0 | 20.0 | 26.5 | 25.5 | 27.0 | 31.0 | 34.2 | 39.5 |
| | Qwen-2.5-Omni-7B (Xu et al., 2025) | 7B | 30.5 | 25.0 | 24.4 | 47.5 | 23.5 | 30.0 | 26.5 | 50.0 | 30.0 | 24.5 | 28.5 | 24.5 | 31.5 | 39.7 | 39.5 |
| | AV-Reasoner (Lu et al., 2025) | 7B | 29.0 | 55.0 | 22.8 | 40.0 | 24.5 | 31.5 | 22.5 | 55.0 | 25.0 | 25.0 | 27.5 | 27.0 | 32.5 | 51.3 | 35.4 |
| Closed Source | Gemini 1.5 Flash (Team et al., 2024a) | - | 23.5 | 40.0 | 21.3 | 31.0 | 27.5 | 32.5 | 28.0 | 30.0 | 30.0 | 27.5 | 23.5 | 25.0 | 28.5 | 27.6 | 34.9 |
| | Gemini 1.5 Flash-8B (Team et al., 2024a) | - | 24.5 | 25.0 | 25.9 | 33.0 | 27.5 | 32.0 | 24.5 | 40.0 | 15.0 | 31.0 | 25.5 | 26.0 | 29.0 | 25.6 | 34.9 |
| | Gemini 1.5 Pro (Team et al., 2024a) | - | **29.5** | 50.0 | 25.4 | 42.5 | **28.0** | 28.5 | **29.0** | 35.0 | 40.0 | 30.0 | 24.5 | 28.5 | 20.5 | 32.2 | 33.8 |
| | Gemini 2.5 Pro (Google, 2025) | - | 38.5 | 55.0 | 27.4 | 79.5 | 24.5 | 50.5 | 27.5 | 45.0 | 20.0 | 18.5 | 19.5 | 33.5 | 40.5 | 38.2 | 37.4 |
| | Reka Core (Team et al., 2024b) | 67B | 28.5 | 20.0 | 22.8 | 24.5 | 27.5 | 30.0 | 25.5 | 25.0 | 20.0 | 30.0 | 25.5 | 24.0 | 24.0 | 33.7 | 34.9 |
| | Reka Flash (Team et al., 2024b) | 21B | 24.5 | 20.0 | 30.5 | 29.5 | 27.5 | 25.5 | 24.5 | 45.0 | 15.0 | 30.0 | 25.5 | 27.0 | 31.5 | 19.1 | 29.2 |
| | Reka Edge (Team et al., 2024b) | 7B | 20.5 | 20.0 | 24.9 | 24.5 | 27.5 | 30.0 | 24.0 | 30.0 | 15.0 | 30.0 | 25.5 | 21.0 | 22.5 | 38.2 | 35.4 |
| | GPT-4o visual caption (Hurst et al., 2024) | - | 26.0 | 55.0 | 24.4 | **48.0** | 27.0 | **34.5** | 23.5 | 25.0 | 30.0 | 21.5 | 22.5 | 32.5 | 23.0 | 32.2 | 25.6 |
| | GPT-4o audio caption (Hurst et al., 2024) | - | 28.0 | **70.0** | 24.4 | **56.5** | 27.5 | 32.5 | 22.5 | 30.0 | 35.0 | 23.5 | 25.5 | **33.5** | 25.0 | 30.2 | 22.0 |

# E CASE STUDY

## LIST OF FIGURES

Figure 9: A sampled error case in the instrument recognition task.

---

❌      **Timbre: Singer Recognition**

**Question:** From the timbre in [audio1], which individual in [img1], [img2], [img3], or [img4] is most likely the singer?
A: the second image
B: the fourth image
C: the third image
D: the first image

[audio1] 🔊     audio content: an AI covered song with Rihanna's vocal timbre

[img1]       [img2]       [img3]       [img4]

**Gemini 1.5 Pro:** The singer in the audio clip is most likely **B: the fourth image**, Billie Eilish. This is based on the distinctive vocal timbre and phrasing, which are characteristic of Eilish's style.

**Error Category:** **Audio Understanding Error**
**Error Reason:** Gemini 1.5 Pro accurately identified the visual content in the image but made an error in audio recognition, incorrectly attributing Rihanna's vocal timbre to Billie Eilish.

**Correct Answer:** D: the first image

Figure 10: A sampled error case in the singer recognition task.

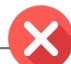

**Timbre: Gunshot Recognition**

**Question:** Which image [img1], [img2], [img3], [img4] do you think best aligns with the gunfire audio [audio1]?
A: the fourth image
B: the third image
C: the second image
D: the first image

[audio1] 🔊 audio content: The sound of the AK-12 firing precise single bursts instead of continuous automatic fire

[img1] 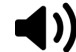 [img2] 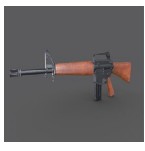 [img3] 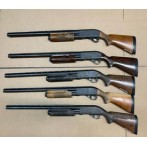 [img4] 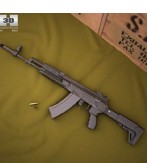

**Gemini 1.5 Pro:** The audio clip sounds like sustained automatic gunfire, which best aligns with the first image, a machine gun. Therefore, the answer is D. The other images depict weapons that are not typically capable of sustained automatic fire in the same way.

**Error Category: Audio Understanding Error**
**Error Reason:** Gemini 1.5 Pro accurately identified the visual content in the images but made an error in audio recognition, incorrectly interpreting the sound of the AK-12's controlled burst fire as sustained automatic gunfire, leading to the wrong conclusion that the answer is D, a machine gun.

**Correct Answer:** A: the fourth image

Figure 11: A sampled error case in the gunshot recognition task.

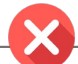

## Timbre: Bird Recognition

**Question:** Which of the four audio segments ([audio1] [audio2] [audio3] [audio4]) shows the bird sound related to the image [img1]?
A: the second audio
B: the third audio
C: the fourth audio
D: the first audio

[audio1] 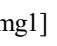  audio content: sound of other bird categories

[audio2]  audio content: sound of other bird categories

[audio3]  audio content: sound of other bird categories

[audio4]  audio content: sound of common grackle birds

[img1]

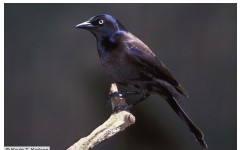

**Gemini 1.5 Pro:** The bird in the image is a Brewer's Blackbird. The second audio clip contains the Brewer's Blackbird's call. So the answer is A.

**Error Category: Audio Understanding Error, Vision Understanding Error**
**Error Reason:** Gemini 1.5 Pro made errors in both visual and audio recognition: it incorrectly identified the common grackle bird in the image as a Brewer's Blackbird and subsequently misattributed the sound of the common grackle to that of a Brewer's Blackbird.

**Correct Answer:** C: the fourth audio

Figure 12: A sampled error case in the bird recognition task.

**Timbre: Animal Recognition**

**Question:** From the four audio options [audio1] [audio2] [audio3] [audio4], can you find the animal audio that corresponds to the image [img1]?
A: the third audio
B: the fourth audio
C: the first audio
D: the second audio

[audio1] 🔊 audio content: cat meow

[audio2] 🔊 audio content: chicken cluck

[audio3] 🔊 audio content: elephant trumpet

[audio4] 🔊 audio content: frog croak

[img1]

**Gemini 1.5 Pro:** The audio of a cat meowing is the fourth audio clip. So the answer is B.

**Error Category: Audio Understanding Error**
**Error Reason:** Gemini 1.5 Pro correctly identified the cat in the image but made an error in audio recognition, mistakenly identifying the sound of a frog as a cat's meow.

**Correct Answer:** C: the first audio

Figure 13: A sampled error case in the animal recognition task.

### ❌ Timbre: Transportation Recognition

**Question:** Select the image ([img1], [img2], [img3], [img4]) that best represents the sound of a vehicle in [audio1].
A: the fourth image
B: the second image
C: the first image
D: the third image

[audio1] 🔊 audio content: the sound of an airplane

[img1] 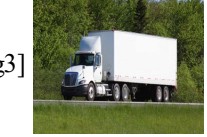 [img2] 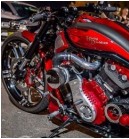 [img3] [img4]

**Gemini 1.5 Pro:** The sound clip is of a motorcycle engine. Therefore, the correct answer is the fourth image, which depicts a Harley-Davidson motorcycle. Final Answer: The final answer is $\\boxed{A}$

**Error Category: Audio Understanding Error**
**Error Reason:** Gemini 1.5 Pro correctly identified the motorcycle in the image but made an error in audio recognition, mistakenly identifying the sound of an airplane as the sound of a motorcycle.

**Correct Answer:** B: the second image

Figure 14: A sampled error case in the transportation recognition task.

## Timbre: Material Recognition

**Question:** Based on [audio1], which of the materials shown in [img1] [img2] [img3] [img4] is most likely to be the source?
A: the second image
B: the first image
C: the third image
D: the fourth image

[audio1] 🔊 audio content: the sound of hitting or stepping on fallen leaves

[img1] [img2] [img3] [img4]

**Gemini 1.5 Pro:** None of these images appear to be the source of the first image, which depicts a wooded path in autumn. The other images show water, tile, and a glass. There's no direct connection between any of them and the forest path.

**Error Category:** Text Understanding Error
**Error Reason:** Gemini 1.5 Pro misunderstood the question text, incorrectly interpreting it as asking which image the first image originates from, rather than identifying which material depicted in the images most likely matches the sound in the audio clip.

**Correct Answer:** B: the first image

Figure 15: A sampled error case in the material recognition task.

**Timbre: Scene Recognition**

**Question:** Choose the picture that you think best fits the audio [audio1] from [img1] [img2] [img3] [img4].
A: the third image
B: the fourth image
C: the first image
D: the second image

[audio1] 🔊 audio content: the sound of traffic street with cars

[img1] [img2] [img3] [img4]

**Gemini 1.5 Pro:** The audio sounds like the whooshing of a train passing quickly. Therefore, the image that best fits the audio is the inside of a subway train (C).

**Error Category: Audio Understanding Error**
**Error Reason:** Gemini 1.5 Pro correctly identified the content of the image but made an error in audio recognition, mistakenly identifying the sound of the traffic street with cars as the sound inside of a subway train.

**Correct Answer:** B: the fourth image

Figure 16: A sampled error case in the scene recognition task.

---

❌ **Timbre: Hazard Recognition**

**Question:** Among [img1], [img2], [img3], and [img4], which potentially dangerous event fits the sound of [audio1] best?
A: the third image
B: the second image
C: the fourth image
D: the first image

[audio1] 🔊 audio content: the sound of fire burning

[img1]    [img2]    [img3]    [img4]

**Gemini 1.5 Pro:** The sound is a high-pitched whistling or hissing, like steam escaping under pressure. This best fits the boiling water in the fourth image. So the answer is C.

**Error Category: Audio Understanding Error**
**Error Reason:** Gemini 1.5 Pro correctly identified the content of the image but made an error in audio recognition, mistakenly identifying the sound of fire burning as the sound of water boiling.

**Correct Answer:** A: the third image

Figure 17: A sampled error case in the hazard recognition task.

Figure 18: A sampled error case in the action recognition task.

---

### ❌ Timbre: Eating Sound Recognition

**Question:** From the foods in the images [img1], [img2], [img3], or [img4], which one is most likely to create the sound [audio1] when consumed?
A: the fourth image
B: the first image
C: the second image
D: the third image

[audio1] 🔊 audio content: the sound of eating juicy grapes

[img1] [img2] [img3] [img4]

**Gemini 1.5 Pro:** The answer is D: the third image (French fries). The sound "crunch" is associated with crispy or brittle foods. French fries, when properly cooked, have a crisp exterior that makes a crunching sound when bitten into. The other foods are less likely to produce this specific sound. Grapes are juicy and soft, chocolate is typically smooth or melts in the mouth, and noodles, though they can have some texture, don't typically produce a distinct crunch.

**Error Category: Audio Understanding Error**
**Error Reason:** Gemini 1.5 Pro correctly identified the content of the image but made an error in audio recognition, mistakenly identifying the sound of eating juicy grapes as the sound of eating crispy chips.

**Correct Answer:** B: the first image

Figure 19: A sampled error case in the eating sound recognition task.

Figure 20: A sampled error case in the speech sentiment analysis task.

**Tone: Meme Understanding**

**Question:** According to [audio1] and [video1], what is funny about this meme?

A: The humor comes from the cat's overly excited face, which contradicts the serious tone of the sound, creating a funny visual and audio mismatch.

B: The meme is funny because the cat's expression is overly dramatic, and the sound is yelled, adding intensity to simple or mundane events.

C: The humor comes from the perfect match between the cat's blank, emotionless stare and the disinterested sound, both of which represent a complete lack of energy or surprise, making even absurd situations seem trivial.

D: It's funny because the cat's face shows anger, but the sound is very calm, making the reaction seem less serious than expected in a high-stress scenario.

[audio1] 🔊 audio content: "bruh"

[video1]

**Gemini 1.5 Pro:** Reject to Answer.

**Error Category: Reject to Answer**
**Error Reason:** Content may be mistakenly flagged for security reasons by Gemini 1.5 Pro, preventing the model from providing an answer.

**Correct Answer:** C

Figure 21: A sampled error case in the meme understanding task.

**Melody: Music Sentiment Analysis**

**Question:** Which image [img1] [img2] [img3] [img4] best represents the emotion conveyed by the music [audio1]?
A: the fourth image
B: the second image
C: the first image
D: the third image

[audio1] 🔊 audio content: upbeat and cheerful music

[img1]     [img2]     [img3]     [img4]

**Gemini 1.5 Pro:** The music seems to evoke sadness and melancholy. Therefore, the best choice is A, the fourth image, which depicts crying emoji faces.

**Error Category: Audio Understanding Error**
**Error Reason:** Gemini 1.5 Pro correctly identified the content of the image but made an error in audio recognition, mistakenly identifying joyful music as sad music.

**Correct Answer:** C: the first image

Figure 22: A sampled error case in the music sentiment analysis task.

Figure 23: A sampled error case in the music genre classification task.

Figure 24: A sampled error case in the dance and music matching task.

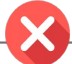

### Melody: Film and Music Matching

**Question:** In your view, which movie scene in the images [img1] [img2] [img3] [img4] do you feel is most suitable for this music [audio1]?
A: the third image
B: the second image
C: the first image
D: the fourth image

[audio1] 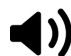  audio content: fast-paced, cheerful music accompanied by comical screams

[img1] 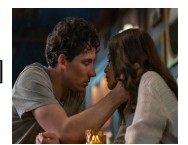  [img2] 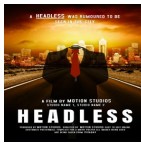  [img3] 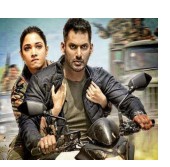  [img4] 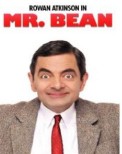

**Gemini 1.5 Pro:** I'd say A, the third image, is the most suitable. The fast-paced, somewhat chaotic energy of the music fits an action scene like the one depicted, with the characters on a motorcycle. The other images suggest romance, thriller, or comedy, none of which are as good a match for this particular piece of music.

**Error Category: Reasoning Error**
**Error Reason:** Gemini 1.5 Pro successfully understood both the audio and visual inputs but made a reasoning error when interpreting the data. It incorrectly associated the fast-paced nature of the music with the action genre, assuming that the combination of high energy and disorder aligns with action films. However, the audio, characterized by fast-paced music accompanied by comical screams, is more indicative of a comedy film rather than an action movie. This mistake highlights a gap in the model's contextual reasoning, as it overlooked the playful and humorous tone of the audio, which is a key attribute of comedy rather than action.

**Correct Answer:** D: the fourth image

Figure 25: A sampled error case in the film and music matching task.

## Melody: Music Score Matching

**Question:** Focusing on the rhythmic phrasing and articulation, which musical score in these images [img1] [img2] [img3] [img4] most closely matches this audio file [audio1]?
A: the third image
B: the second image
C: the fourth image
D: the first image

[audio1] 🔊 audio content: slow-paced music with a long, sustained vocal at the end

[img1]    [img2]    [img3]    [img4]

**Gemini 1.5 Pro:** The answer is C: the fourth image. The audio clip has a moderately paced, syncopated rhythm with a swing feel. The fourth image features eighth and sixteenth notes grouped in a way that creates this rhythmic character, along with staccato markings that suggest a short, detached articulation similar to what's heard in the audio. The other images either have a slower tempo, longer sustained notes, or different rhythmic groupings that don't match the audio.

**Error Category: Audio Understanding Error, Reasoning Error**
**Error Reason:** First, an audio understanding error occurred, as it failed to accurately interpret the audio input, which features slow-paced music with a long, sustained vocal at the end. Instead, the model incorrectly identified the audio as having a moderately paced, syncopated rhythm with a swing feel, entirely mischaracterizing its tempo and articulation. Second, the model incorrectly concluded that the fourth image, with its eighth and sixteenth notes and staccato markings, matches the rhythmic character. However, staccato markings, indicating short, detached articulation, conflict with the described swing feel, which requires smoother, connected phrasing. This reliance on rhythm alone is limiting, as pitch is equally important in musical notation. The model needs to capture the pitch in the audio and align it with the notes in the images to make accurate judgments.

**Correct Answer:** B: the second image

Figure 26: A sampled error case in the music score matching task.

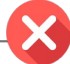

## Space: Audio 3D Angle Estimation

**Question:** Based on [audio1], could you provide the azimuth and elevation angles of the sound located in front of the man dressed in white and black, as illustrated in [img1]? The angles are measured egocentrically, with the azimuth ranging from -180° to 180° (left to right) and the elevation ranging from -90° to 90° (bottom to top).
A: azimuth: 0°, elevation: -1°
B: azimuth: 38°, elevation: 9°
C: azimuth: 51°, elevation: -6°
D: azimuth: 54°, elevation: 10°

[audio1] 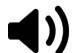 audio content: 4-channel spatial audio featuring men's voice and the sound of water flowing from a tap

[img1] 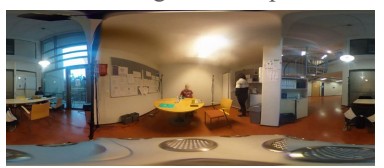

**Gemini 1.5 Pro:** The man in the maroon shirt is the one dressed in white and black. The microphone pointed at him appears to be at approximately 0° azimuth (directly in front) and about -1° elevation (slightly below the horizontal). So the answer is A.

**Error Category: Vision Understanding Error, Audio Understanding Error**
**Error Reason:** First, the statement, "The man in the maroon shirt is the one dressed in white and black," is incorrect. The individual dressed in white and black is not the same as the man in the maroon shirt but is located elsewhere in the scene. This indicates a failure to accurately identify and associate visual details, leading to a misunderstanding of the relationship between these individuals.
Second, It failed to identify the sound of water flowing from the tap, which is visually obscured behind the individual dressed in white and black. This highlights a critical limitation in linking audio cues to visually occluded sources. Additionally, the model was unable to accurately determine the 3D spatial orientation of the tap's water flow.

**Correct Answer:** C: azimuth: 51°, elevation: -6°

Figure 27: A sampled error case in the audio 3D angle estimation task.

---

### ❌ Space: Audio Distance Estimation

**Question:** Using [audio1] as a reference, could you determine the distance of the sound coming from below the man in [img1]? The distance should be measured egocentrically in centimeters.
A: 143
B: 137
C: 115
D: 109

[audio1] 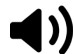 audio content: 4-channel spatial audio featuring people's talking, as well as the sound of a vacuum cleaner

[img1]

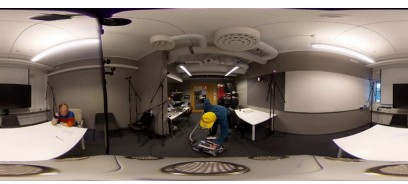

**Gemini 1.5 Pro:** The sound source is the vacuum cleaner being operated by the person in the yellow hat. Considering the camera's position (likely near the head of the person sitting at the table), the vacuum is roughly 115 centimeters away. So the answer is **C: 115**.

**Error Category: Audio Understanding Error**
**Error Reason:** Gemini 1.5 Pro correctly recognized the visual content in the scene but failed to effectively utilize the spatial audio data to determine the precise distance of the vacuum cleaner. While the audio content featured 4-channel spatial information that could have been used to calculate the vacuum cleaner's exact position relative to the camera, the model instead relied exclusively on visual estimations of the distance. This reliance on visual cues alone ignored the valuable depth and directional data provided by the audio, resulting in an inaccurate calculation of 115 centimeters. The error underscores a critical gap in multi-modal reasoning, where the model did not effectively integrate spatial audio with visual context. This highlights the need for enhanced alignment between auditory and visual processing, especially in scenarios where audio provides essential spatial information that visual inputs alone cannot resolve accurately.

**Correct Answer:** B: 137

Figure 28: A sampled error case in the audio distance estimation task.

Figure 29: A sampled error case in the audio time estimation task.

**Time: Audio-Visual Synchronization**

**Question:** Can you identify which of [audio1], [audio2], [audio3], or [audio4] best synchronizes with [video1]?
A: the first audio
B: the third audio
C: the fourth audio
D: the second audio

[audio1] audio content: audio with random offsets, speed-ups, slow-downs

[audio2] audio content: audio with random offsets, speed-ups, slow-downs

[audio3] audio content: the original audio corresponding to [video1]

[audio4] audio content: audio with random offsets, speed-ups, slow-downs

[video1]

**Gemini 1.5 Pro:** The first audio best synchronizes with the video. The correct answer is A.

**Error Category: Audio Understanding Error, Vision Understanding Error**
**Error Reason:** Gemini 1.5 Pro failed to understand the temporal synchronization between visual and audio elements, leading to errors in aligning events across modalities.

**Correct Answer:** B: the third audio

Figure 30: A sampled error case in the audio-visual synchronization task.

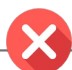

### Time: Action Sequencing

**Question:** Based on [audio1], which order most likely represents the actions in [img1], [img2], [img3], and [img4]?
A: the first image --> the second image --> the third image --> the fourth image
B: the third image --> the fourth image --> the second image --> the first image
C: the first image --> the fourth image --> the third image --> the second image
D: the third image --> the fourth image --> the first image --> the second image

[audio1] 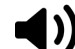  audio content: the sound of a series of actions

[img1] 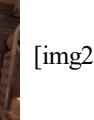  [img2] 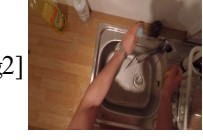  [img3] 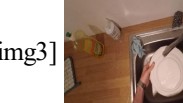  [img4] 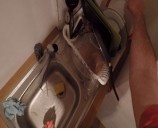

**Gemini 1.5 Pro:** The correct answer is A. Here's why: **Image 1:** Shows a potato being washed under the tap. This is likely the first step, as raw food preparation usually precedes dishwashing. **Image 2:** Shows a sponge and an empty sink with running water, implying the potato has been washed and now the dishes are being cleaned.* **Image 3:** Shows a plate being washed. This follows the cleaning of the potato and the preparation of the sink with soapy water. **Image 4:** Shows the plate being placed in a drying rack with other clean dishes, indicating the dishwashing process is nearing completion. The other options don't make logical sense in terms of the typical sequence of washing a potato and then doing dishes.

**Error Category: Audio Understanding Error, Reasoning Error**
**Error Reason:** Gemini 1.5 Pro failed to accurately identify the sequence of actions based on the audio cues. The model misinterpreted the order in which sounds corresponding to specific actions occurred, leading to an incorrect understanding of the temporal flow of events. Furthermore, it compounded this error with a reasoning mistake by incorrectly inferring the logical order of the actions.

**Correct Answer:** D: the third image --> the fourth image --> the first image --> the second image

Figure 31: A sampled error case in the action sequencing task.

Figure 32: A sampled error case in the hallucination evaluation task.

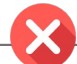

## Intricacy: Action Prediction

**Question:** With [img1] preceding [audio1], what does the audio suggest about this person's actions?
A: turn box
B: wrap up coffee
C: smell box
D: put coffee in cupboard

[audio1] 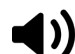 audio content: the sound of actions

[img1]

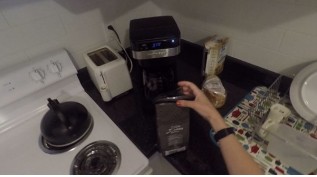

**Gemini 1.5 Pro:** The audio suggests the person is opening a box or bag, most likely of coffee. Therefore the answer is **B: wrap up coffee**.

**Error Category: Audio Understanding Error, Reasoning Error**
**Error Reason:** Gemini 1.5 Pro failed to correctly understand the audio, leading to a misinterpretation of the events. Additionally, the model was unable to understand the order of actions, resulting in an incorrect ordering of the events. This highlights deficiencies in both audio recognition and the ability to reason about temporal relationships between actions.

**Correct Answer:** D: put coffee in cupboard

Figure 33: A sampled error case in the action prediction task.

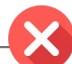

**Intricacy: Action Tracing**

**Question:** With [img1] and the preceding [audio1] in mind, what actions were being performed by this person during the audio?
A: bend machine
B: rinse chopping board
C: carry machine
D: wipe chopping board with rag

[audio1] 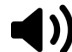   audio content: the sound of actions

[img1]

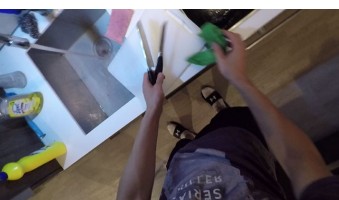

**Gemini 1.5 Pro:** The person is wiping the chopping board with a rag (D).

**Error Category: Text Understanding Error**
**Error Reason:** Gemini 1.5 Pro misinterpreted the question, failing to recognize the temporal focus of the query. The model incorrectly assumed the question was asking about the person's current actions, rather than the actions at the time the audio occurred. This misunderstanding of both the question's intent and its temporal context led to an incorrect response.

**Correct Answer:** B: rinse chopping board

Figure 34: A sampled error case in the action tracing task.

