# OpenReview forum: "AV-Odyssey Bench: From Fundamental Audio Perception to Audio-Visual Understanding"
_ICLR.cc/2026/Conference — ICLR 2026 Conference Withdrawn Submission_

### Official Review · Reviewer_Joa2 · 2025-10-31

**Soundness:** 2
**Presentation:** 1
**Contribution:** 2
**Rating:** 2
**Confidence:** 4

**Summary:**

This paper proposes a new audio–visual benchmark, AV-Odyssey Bench, designed to evaluate whether multimodal large language models (MLLMs) can unify cross-modal reasoning by leveraging synchronized audio–visual cues to infer solutions. Prior to this, the authors design DeafTest to demonstrate that MLLMs struggle with basic auditory perception. This weak auditory perception is found to correlate with performance on AV-Odyssey, emphasizing that effective audio–visual comprehension requires attention to fundamental audio perception. The proposed benchmark, AV-Odyssey, contains 4,555 QA pairs spanning 26 different tasks that require both audio and visual signals for reasoning. Experimental results show that both open-source and closed-source models perform poorly on AV-Odyssey, highlighting significant room for improvement in MLLMs’ audio–visual understanding capabilities.

**Strengths:**

- Although several audio–visual understanding benchmarks already exist, proposing a new benchmark with 26 diverse tasks that require both audio and visual modalities is valuable to the community.

- Designing a preliminary diagnostic experiment (DeafTest) is an interesting and meaningful way to assess the perceptual capabilities of current MLLMs.

**Weaknesses:**

- Although the benchmark is interesting, some tasks seem to evaluate incidental abilities of MLLMs rather than general audio–visual understanding.
  - While expert-level knowledge might make certain tasks challenging, their inclusion is questionable given the limited size of the benchmark (4,555 QA pairs).
  - For example, the interleaved text–audio–image example shown in Figure 3 makes the dataset seem unreliable. If the model—or even a human—does not know who Taylor Swift is, then the question becomes ambiguous, as all options are female singers, leaving no definite answer. If several QA pairs are constructed in this way, the benchmark may not serve as a fair and meaningful evaluation.
  - Similarly, in the Timbre tasks (L 314), the authors mention questions involving singer recognition or bird-species identification, which again depend on specific knowledge rather than general multimodal reasoning.
  - For the Melody tasks involving dance–music matching, it is unclear whether this is an important evaluation for MLLMs. Can humans also perform reliably on such tasks, and are there definitive answers?
  - For Space tasks, do models process multichannel audio, or are the signals collapsed to mono? If it is the latter, then such tasks may be unsolvable for the models.
  - In Figure 5, musical instrument recognition is categorized as a hallucination evaluation, but it seems more like a straightforward instrument-sound detection task.
  - Given that both open- and closed-source models perform poorly in Table 4, does this indicate that the benchmark may be overly difficult? How would human participants perform on the same benchmark?
 - If the dataset were larger, these specialized QA types could be more informative. However, with only 4,555 QA pairs, expert-level or knowledge-based questions may not contribute meaningfully to assessing general audio–visual reasoning.

- Lack of dataset construction details, which makes the benchmark’s reliability questionable.
  - Which data sources (audio and video) were used to build DeafTest and AV-Odyssey? Were they newly collected or repurposed from existing datasets?
  - How were the question–answer pairs created? Were they manually written or generated with LLM assistance? How were negative samples selected for multiple-choice QAs?
  - Was any human verification performed for quality control and validation? If not, how can the authors assure that this dataset is accurate enough?

- Insufficient details on experimental design
  - In Table 5, only 300 randomly selected questions were evaluated. Why not evaluate on the entire benchmark? Were these 300 samples uniformly drawn across all tasks?
  - Also in Table 5, how were ground-truth audio captions obtained for all audio samples? Were these annotated by humans? For instance, for tasks like audio distance estimation or music sentiment analysis, what do these captions look like? Examples would help clarify.

- Poor visualization quality
  - Several figures and tables are difficult to read.
  - For example, the text in Figures 7 and 8 is too small to interpret clearly.

**Questions:**

- In many audio model implementations, input audio is normalized to a consistent scale before being fed to the model. For the loudness comparison in Table 1, do the authors think the poor performance results from the model’s inability to differentiate loudness, or could normalization have affected the results?

- In L 368, adding a line break before Prompt Design would improve readability.

- What potential future research directions could enhance audio–visual perception and reasoning in MLLMs?

---

### Official Review · Reviewer_7k45 · 2025-11-01

**Soundness:** 4
**Presentation:** 4
**Contribution:** 4
**Rating:** 8
**Confidence:** 5

**Summary:**

This paper introduces AV-Odyssey Bench, a comprehensive benchmark designed to evaluate MLLMs across both low-level audio perception and high-level audio-visual reasoning. The work stems from an important empirical observation that state-of-the-art models often fail at basic auditory perception tasks such as loudness or pitch comparison, despite performing well on complex multimodal reasoning. To systematically diagnose this gap, the authors propose two complementary components: a) DeafTest – a suite of four fundamental audio perception tasks (sound counting, loudness, pitch, and duration discrimination). b) AV-Odyssey Bench – a large-scale benchmark with 4,555 multiple-choice questions spanning 26 tasks and 10 domains (e.g., music, daily life, animals, memes, spatial, and temporal reasoning). Tasks involve interleaved audio, image, video, and text inputs to evaluate true multimodal integration rather than unimodal shortcuts. The central hypothesis is that deficiencies in low-level auditory perception constrain high-level multimodal reasoning.

**Strengths:**

- Novel approach connecting perceptual (DeafTest) and reasoning (AV-Odyssey) capabilities.
- 26 tasks across timbre, tone, melody, space, time, hallucination, and intricacy dimensions.
- Objective, reproducible evaluation via multiple-choice format—avoiding LLM-based grading biases.
- One of the first papers that demonstrates that “audio perception” is the missing foundation for audio-visual intelligence.
- Also has multi-audio comparison instances, which is nice.

**Weaknesses:**

- Human baseline is already at 90%+, which is good (means the model cannot solve simpler questions) and bad (the questions are actually not very difficult).
- The instances in the final benchmark are still short (as far as I understand).
- Not too many weaknesses. Well-rounded paper.

**Questions:**

- What is the impact of not understanding surface level audio properties on other benchmarks? If I understand, is it the case that most current AV benchmarks only assess semantic understanding?

---

### Official Review · Reviewer_4Ydp · 2025-11-01

**Soundness:** 2
**Presentation:** 3
**Contribution:** 2
**Rating:** 2
**Confidence:** 4

**Summary:**

This paper presents AV-Odyssey Bench, a comprehensive benchmark designed to evaluate audio-visual reasoning in multimodal large language models (MLLMs).
The authors introduce DeafTest, a suite of four basic auditory perception tasks, including sound counting, loudness, pitch, and duration, which reveal that most leading MLLMs, such as GPT-4o, Gemini, and Reka, perform close to random guessing, except Gemini 2.5 Pro.

Building on this, AV-Odyssey expands the evaluation to more complex multimodal challenges involving timbre, tone, melody, spatial and temporal reasoning, hallucination detection, and intricate cross-modal understanding, all formatted as multiple-choice questions for objective scoring.
Experimental results indicate that current models perform only marginally above chance, underscoring their limited capacity for cross-modal integration.

Although DeafTest and AV-Odyssey target different levels of analysis, the study finds a strong correlation between their results, suggesting that deficits in low-level auditory perception significantly limit higher-order multimodal reasoning.

**Strengths:**

- Introduces the two benchmarks (DeafTest + AV-Odyssey) explicitly linking low-level auditory perception to high-level audio-visual reasoning.

- The proposed benchmark is comprehensive, covering diverse sound categories (environmental sounds, music) and audio-visual scenarios, and identifying weaknesses in the auditory perception of MLLMs as a bottleneck is an important finding

- Correlation analysis demonstrates that fundamental auditory deficits cascade into multimodal reasoning failures.

**Weaknesses:**

- When first reading this submission, this reviewer expected to see which parts are attributed to the failure of the audio-visual integration, e.g., audio encoder, audio token mapping, training dataset, LLM, and its decoder. Clearly identifying the causes would directly provide the guidance for which part should be further improved. However, the current version does not provide any detailed analysis to reveal more specific causes. While the findings obtained from the correlation between DeafTest and AV-Odyssey are interesting, the role and message of this work are limited in this sense.

- Likewise, the overall depth of analysis is lacking.

- While auditory weaknesses are emphasized, visual perception and fusion mechanisms receive less analytic attention. As shown in Fig. 5, many tests are also related to the visual perception and multi-modal association. Since the visual perception capability is also not perfect, when the audio-visual reasoning capability is assessed, the imperfection of the visual perception should be considered. Thereby, the effect of the audio capability can be clearly analyzed in a more isolated way.

- It would significantly improve the impact of the submission if the authors also propose a quick remedy or a way to improve the current bottleneck based on the findings in this work.

- Human baseline missing. Direct human evaluation on AV-Odyssey tasks would better quantify the human-model gap.

- While it was interesting to learn the correlation between DeafTest and AV-Odyssey from this work, this reviewer feels some disconnection between the two datasets.

**Questions:**

- What was the design decision about the task selection in Fig. 5?
Do those span most of the real-world scenarios? What is the design motivation to select those tasks?

---

### Official Review · Reviewer_zC5Z · 2025-11-11

**Soundness:** 3
**Presentation:** 2
**Contribution:** 2
**Rating:** 4
**Confidence:** 4

**Summary:**

An Audio Visual Benchmark that focuses on showing audio perception has fundamental to AudioVisual Understanding.
The paper shows that a hearing test predicts well the performance on the benchmark.

**Strengths:**

The importance of hearing for audio visual perception is well brought out. The tests for hearing are well thought out -- counting, loudness,

**Weaknesses:**

The paper is pursuing a couple of different ideas -- building an audiovisual benchmark, and showing that hearing is important for performance. In this pursuit, both objectives are not well served. In particular, it is not clear how well the visual understanding was studied by this benchmark

**Questions:**

I would have liked to see an evaluation of just large audio language models on the hearing test.

---

### Note · Authors · 2025-11-21

I have read and agree with the venue's withdrawal policy on behalf of myself and my co-authors.